# Molecular Biomarkers in Perthes Disease: A Review

**DOI:** 10.3390/diagnostics13030471

**Published:** 2023-01-27

**Authors:** Vesna Spasovski, Sanja Srzentić Dražilov, Gordana Nikčević, Zoran Baščarević, Maja Stojiljković, Sonja Pavlović, Duško Spasovski

**Affiliations:** 1Institute of Molecular Genetics and Genetic Engineering, University of Belgrade, Vojvode Stepe 444a, 11010 Belgrade, Serbia; 2School of Medicine, University of Belgrade, Dr Subotica 8, 11000 Belgrade, Serbia; 3Institute of Orthopedics “Banjica”, Mihajla Avramovića 28, 11000 Belgrade, Serbia

**Keywords:** Perthes disease, candidate genes, biomarkers, biological therapy

## Abstract

Background: Perthes disease is a juvenile form of osteonecrosis of the femoral head that affects children under the age of 15. One hundred years after its discovery, some light has been shed on its etiology and the biological factors relevant to its etiology and disease severity. Methods: The aim of this study was to summarize the literature findings on the biological factors relevant to the pathogenesis of Perthes disease, their diagnostic and clinical significance, and their therapeutic potential. A special focus on candidate genes as susceptibility factors and factors relevant to clinical severity was made, where studies reporting clinical or preclinical results were considered as the inclusion criteria. PubMed databases were searched by two independent researchers. Sixty-eight articles were included in this review. Results on the factors relevant to vascular involvement and inflammatory molecules indicated as factors that contribute to impaired bone remodeling have been summarized. Moreover, several candidate genes relevant to an active phase of the disease have been suggested as possible biological therapeutic targets. Conclusions: Delineation of molecular biomarkers that underlie the pathophysiological process of Perthes disease can allow for the provision of earlier and more accurate diagnoses of the disease and more precise follow-ups and treatment in the early phases of the disease.

## 1. Introduction

Perthes disease (LCPD OMIM#150600) is an idiopathic, debilitating structural disorder that affects the growing femoral head of children, mostly boys. The male-to-female ratio is reported to be 3.1:1 [1]. Annual incidence is highly variable between populations and, according to 21 studies from 16 countries, the incidence rate was found to range from 0.2 incidences per 100,000 people to 19.1 incidences per 100,000 people [2].

It is generally accepted that the main hallmark of Perthes disease is the disruption of the blood supply to the femoral capital physis, which can lead to the collapse of the femoral head, loss of sphericity and subsequent degenerative changes in the joint and surrounding tissue. The natural course of the disease goes through four phases (hip synovitis, femoral epiphysis condensation and fragmentation, reossification, and remodeling) that last several years, eventually causing permanent mechanical disturbance of the joint in varying degrees of intensity. Research conducted over a period of more than 100 years included external factors connected with socioeconomic status, such as low birth weight and the mother’s smoking habits, and other factors such as obesity and hyperactivity. The elucidation of the etiology of the disease from this research brought many insights, though a definite cause is still unknown [3]. However, the results of more recent research that explored biological and physiological mechanisms have given many new answers, and we are one step closer to a better understanding of the disease’s pathogenesis and more accurate treatment for this disabling disease.

Partial- or full-weight bearing restrictions and operative containment methods (femoral and/or pelvic osteotomies) are still considered standard treatment options and are in broad use worldwide [3,4,5]. During the last decade, a class of new therapeutics based on biomarkers has been introduced and their biological potential has been explored in animal studies. This biological therapy provides a possibility to resolve the disease in its initial phase and prevent subsequent complications, thus diminishing the need for operative procedures.

This review summarizes the results from research on biomarkers that are involved in endothelial activation and dysfunction that could be involved in impaired blood supply to the femoral head in Perthes disease. The findings of studies that examined markers relevant to angiogenesis, inflammation and apoptosis have been introduced. An overview of the familial and sporadic occurrence of collagen mutations is presented. We also discuss the possible usage of new techniques for better diagnostics of the disease and the development of biomarker-based therapeutics that directly target the cause of the disease and could be effective in the early phases of the disease, precluding the femoral head collapse of an affected hip.

## 2. Materials and Methods

### Literature Search Strategy and Selection Criteria

The inclusion criteria for this narrative review encompassed available studies that examined factors influencing the development of Perthes disease. This included hereditary factors that could act as susceptibility factors; factors that could contribute to vascular impairment, including markers of angiogenesis, inflammation and apoptosis; and inflammatory biomarkers relevant for clinical presentation. Studies describing biological therapeutics used in preclinical models of Perthes disease were also included. Studies of any level of evidence published in peer review journals were further analyzed.

Exclusion criteria included non-English written articles, conference presentations and expert opinions. Articles without an accessible abstract were also excluded.

Online databases PubMed and Science Direct were searched by two independent authors. The searches were conducted using the following keywords: “Perthes” or “avascular necrosis of femoral head in children” combined with the terms “biomarkers”, “endothelial”, “inflammatory”, “collagen mutations”, “susceptibility”, “hereditary”, “inflammation”, “angiogenesis”, “vascular”, “apoptosis”, “biological therapy” and “genetic factors” in order to gain the most specific results. A time range was set from 1 January 1992. to 1 October 2022. All incongruities were discussed with senior investigators until a consensus was reached.

## 3. Results and Discussion

### 3.1. Endothelial Dysfunction in Perthes Disease

Vascular involvement in the etiology of Perthes disease has been proposed since its discovery and studied in epidemiological and experimental studies. The first genetic markers to be studied in Perthes disease were markers of thrombophilia, and their significance is described elsewhere [3,6]. In this review, we highlight endothelial markers that could be relevant to the vascular dysfunction observed in Perthes patients.

The basic building block of blood vessels is the endothelial cells (ECs). They form a permeability barrier, thus playing a primary role in the maintenance of hemostasis and blood fluidity, preventing platelet aggregation and thrombosis. They contribute to the retention of vasomotor tone, play a decisive role in angiogenesis and vasculogenesis, and are involved in immune and inflammatory responses. They also represent a multifunctional paracrine and endocrine organ [7]. For this reason, endothelial cell injury, activation and/or dysfunction play an important role in the pathogenesis of diseases that involve thrombosis, angiogenesis, vascular repair and/or inflammation, which is the case with Perthes disease.

Dysfunction of the vascular system and its life-long consequences were analyzed by Hailer et al. in a study that involved 3141 patients and 15,595 individuals without Perthes disease who were selected from the Swedish Inpatient Register [8]. The study examined Perthes patients’ risk for vascular diseases during life and included subjects older than 30 years of age at the follow-up. Patients with Perthes disease were found to have significantly higher risks for blood diseases, including anemias, coagulation defects and hypertension, than the control group, implying the involvement of the vascular system in Perthes disease [8].

Anatomical differences in Perthes patients were examined in a study where the size of blood vessels was measured using the technique of flow-mediated dilation of the brachial artery in response to an ischemic stimulus [9]. A notable reduction in blood velocity and blood flow was detected in children with Perthes disease compared to controls, which was caused by the smaller artery caliber found in affected children [9].

#### 3.1.1. VEGF

Vascular endothelial growth factors (VEGFs) are a family of signal peptides, with VEGF-A as the founding member [10]. VEGF-A is an extracellular signal protein that is produced by many cells with the main function of stimulating the formation of blood vessels during embryonic development and in adults. Angiogenesis is a process that is essential during pregnancy and in tissue growth and repair but also in disease development and progression. Among many other functions, VEGF-A acts as a key regulator of endochondral ossification [11] and, therefore, represents a potential biomarker for Perthes disease (Table 1).

Several studies explored the significance of VEGF for Perthes disease. Experimental studies on piglet and rabbit models of Perthes disease explored its role in cartilage remodeling and ossification [12,13]. Kim et al. investigated the role of VEGF in restoring endochondral ossification in the epiphyseal cartilage after ischemic necrosis in a piglet model of Perthes disease [12]. The results of this study showed ectopic expression of VEGF in the proliferative zone of the epiphyseal cartilage, with VEGF mRNA expression upregulation as early as 24 h after the induction of ischemia. This study proposes a possible role of VEGF in osteoclast recruitment and differentiation and the resorption of the necrotic cartilage and bone and suggests that VEGF is a factor that induces the restoration of endochondral ossification in the epiphyseal cartilage through the coupling of angiogenesis, cartilage remodeling and ossification after ischemic damage [12].

Similar findings came from a study on rabbits which showed increased VEGF mRNA and protein expression in the proliferative zone of epiphyseal cartilage and downregulation of VEGF in the hypertrophic zone of epiphyseal cartilage after necrosis [13].

Two studies explored serum levels of VEGF in the active phase of Perthes disease. In a Turkish study conducted by Sezgin et al., serum VEGF levels were compared between 28 patients and 25 controls. There was no significant difference in serum VEGF levels between the two groups [14]. Nevertheless, a study on Indian children showed that median serum levels of VEGF were significantly lower in the patient group compared to the normal, healthy, control group [15]. The study also examined the level of VEGF at four different stages of the disease but failed to find any significant differences. If future studies confirm the significant role of VEGF, especially if differences between various stages of the disease are established, the serum VEGF level may act as a marker for the follow-up of the disease and a possible therapeutic target for Perthes disease.

#### 3.1.2. eNOS

Among markers that could influence endothelial cell fate in the context of osteonecrosis of the femoral head, the endothelial nitric oxide synthase gene (eNOS) has been much investigated. eNOS is primarily responsible for the generation of NO in the vascular endothelium [16]. It is involved in a variety of physiologic processes, including angiogenesis, thrombosis and bone turnover, making it a promising marker for Perthes disease.

There are three polymorphisms of the eNOS gene that have been identified thus far, including a −786 T > C (rs2070744) polymorphism in the promoter region, an 894 G > T (rs1799983) polymorphism in exon 7 and a variable number of tandem 4a4b repeats (rs61722009) in intron 4 [17,18]. A Chinese study that examined 4a4b polymorphism in intron 4 and 894 G > T polymorphism in exon 7 found that both polymorphisms may be risk factors for Perthes disease [19]. An Iranian study that examined all three polymorphisms of eNOS highlighted the association of 894 G > T and −786 T > C but not the 4a4b polymorphism with susceptibility to Perthes disease [20].

#### 3.1.3. Plasma Exosomes and Microparticles

The number of endothelial progenitor cells in circulation represents a valuable tool for the justification of the state of the endothelium and might also be explored in the context of physiological and pathological events. Additionally, circulating endothelial microparticles (EMPs) and exosomes, which represent extracellular vesicles secreted by endothelial cells, may vary in number and content during normal and pathological conditions, and could have clinical significance. The involvement of endothelial cells in the development of Perthes disease was recently studied by the analysis of plasma extracellular vesicles [21]. The study showed an increased number of EMPs in the circulation of Perthes patients and suggested that the mechanism of action of these circulating EMPs might be endothelial cell apoptosis, endothelial dysfunction and deregulated angiogenesis in Perthes patients [21]. In a study by Huang et al., epigenetic factors that might contribute to Perthes disease were examined in the plasma exosomes of both Perthes patients and controls [22]. They showed the differential expression of several miRNAs involved in endothelial cell dysfunction (miR-3133, miR-4644, miR-4693-3p and miR-4693-5p) and osteoclastogenesis (miR-3133, miR-4693-3p, miR-4693-5p, miR-141-3p and miR-30a) that might contribute to the development of Perthes disease [22].

### 3.2. Inflammatory Factors in Perthes Disease

Inflammation plays a fundamental role in Perthes disease as significant and persistent synovitis is present throughout the course of the disease [23]. Synovitis causes cartilage edema, the deterioration of cartilage mechanical properties, cartilage hypermetabolism and cartilage hypertrophy. It is the primary cause of hip pain in affected children. Synovitis has a negative impact on bone formation during the repair process and is involved in the subsequent ischemic osteonecrosis present in Perthes patients. During the fragmentation phase of the disease, an unsuccessful attempt of the body to repair brittle bone tissue of the collapsing femoral head occurs; instead of forming normal bone matter with high strength and low compressibility, a fibrovascular tissue with fragments of necrotic bone is formed [24]. Yet, only in recent years have researchers gained significant data about the nature of the inflammatory process in Perthes disease.

The process of bone turnover represents a fine balance between bone formation and bone resorption, which is based on the activity of two major cell populations in the bone: osteoclasts (OCs) and osteoblasts (OBs) [25,26,27]. It is known that IL-6 positively regulates OCs [28,29] and negatively regulates OBs [29,30]. Its elevation could trigger the uncoupling of bone resorption and bone formation, which is characteristic of this stage of the disease and leads to ischemic osteonecrosis of the developing hip.

The significance of genetic markers of inflammation for the development and activity of Perthes disease, specifically IL-6, TLR4, TNF-α and IL-3, were first studied in associative studies [31,32]. Two promoter polymorphisms in IL-6, 174 G > C and −597 G > A, and two SNVs that encode mutations in the ectodomain of TLR4, D299G and T399I, were examined [31]. The significance of TLR4 was not shown, but heterozygous carriers of promoter variants *IL-6* −174 G > C and −597 G > A had a lower chance of developing Perthes disease than carriers of both homozygote genotypes [31]. Next, variants in TNF-α −308 G > A and IL-3 −16 C > T were analyzed [32]. TNF-α has been recognized as a skeletal catabolic agent that stimulates osteoclastogenesis while simultaneously inhibiting osteoblast function [33], and gene variants in IL-3 were shown to be associated with rheumatoid arthritis [34]. No significant differences between the frequencies of analyzed gene variants in TNF-α or IL-3 genes between the patient and control groups were found [32]. An Iranian study from 2019 examined the significance of *IL-6* −174 G  >  C and −572 G  >  C genotypes in developing Perthes disease in Iranian children [35]. The study showed that the mutant homozygote genotype CC of IL-6 −174 G > C polymorphism was associated with an increased risk of Perthes disease in the Iranian population.

Several studies examined the expression of inflammatory cytokines in the serum of Perthes patients and animal models of the disease.

In a study by Kamiya et al., 27 inflammatory cytokines and related factors were analyzed in the synovial fluid of 13 Perthes patients and significant overexpression of IL-6 was observed [36]. In addition, this study showed that in the active stage of Perthes disease IL-6 is a key inflammatory cytokine present in the synovial fluid, with strong impact on bone turnover. The detailed mechanism of action of IL-6 was studied by Yamaguchi et al., and the relevance of HIF-1α in the activation of IL-6 was designated [23]. Namely, when osteonecrotic articular chondrocytes were cultured under hypoxic conditions, a significant increase in the expression of *IL-1β* and *TNF-α* genes was observed, which was inhibited by the IL-6 receptor blocker tocilizumab. This work highlighted tocilizumab as a potential therapeutic in the treatment of synovitis.

A murine model of Perthes disease was used to further delineate the role of IL-6 in the process of revascularization and new bone formation following ischemic osteonecrosis [37]. Ischemic osteonecrosis was surgically induced in wild-type and IL-6 knockout (KO) mice. Significant differences were observed in OB and OC numbers and bone formation and mineral apposition rates in the osteonecrotic side of the IL-6 KO mice compared to the wild-type mice.

#### TLR4

In order to delineate the mechanism of the inflammatory cascade responsible for the disturbed process of bone remodeling and the receptor mechanisms involved in sensing the necrotic bone observed in this phase, the role of macrophages was studied in detail in a piglet model of Perthes disease [38]. The study showed that macrophages that are exposed to soluble factors from the necrotic bone were activated (M1 phenotype) and showed increased secretion of proinflammatory cytokines TNF-α, IL1-β and IL-6. The upregulation of pattern recognition receptor TLR4 in activated M1 macrophages was determined and the relevance of the TLR4 signaling pathway for proliferation, migration and proinflammatory cytokine responses in activated macrophages was established [38]. It was suggested that the process of inflammation, driven by proinflammatory mediators, could be responsible for the distortion of the regenerative process, where instead of new bone formation, the repair process aids the formation of fibrotic tissue. The relevance of the TLR4 signaling pathway in this process was ultimately confirmed by knockdown experiments using the CRSPR-Cas9 approach and the TLR4 inhibitor, TAK242 [38]. TLR4 is a known receptor where both infectious and non-infectious stimuli converge to elicit a pro-inflammatory response. It is involved in the lipopolysaccharide sensing of gram-negative bacteria and binds the endogenous molecules produced as a result of tissue injury [39]. Hence, these results illuminated a new paradigm for therapy that could be based on TLR4 inhibitors. Since the TLR4 signaling pathway regulates OB and OC formation [40], the damage caused by its misregulation could be repaired using such an approach.

### 3.3. Role of Apoptotic Factors in Uncoupled Bone Metabolism

Several studies explored the mechanism of programmed cell death in the avascular necrosis of the femoral head (ANFH) and reported that the processes leading to the death of femoral head cells involve an increased rate of apoptosis rather than solely bone cell necrosis [32,41,42,43]. The importance of the inner apoptotic pathway was also observed in the peripheral blood mononuclear cells of Perthes patients, where the upregulation of *BAX* gene expression levels and an increased *BAX*/*BCL2* ratio were first shown by Srzentic et al. [32]. Results from a recent study by Zhu et al. revealed elevated levels of *BAX* mRNA and protein in the cartilage, serum and chondrocytes of Perthes patients compared with a healthy control group. Moreover, this study examined the underlying mechanism of the elevation of *BAX* gene expression and showed that miRNA-214 is a direct regulator of the BAX gene [42]. It showed that the downregulation of miR-214 promotes chondrocyte viability and decreases apoptosis via the downregulation of BAX. Taken together, these results indicated that the expression levels of miR-214 and *BAX* might represent reliable biomarkers and potential therapeutic targets in the treatment of Perthes disease.

### 3.4. Familial Clustering and Tween Studies of Perthes Disease

Since the first mentions of Perthes disease described among siblings [2,3], familial cases have been occasionally reported, but statistical analysis usually negates the influence of genetic factors on the genesis of Perthes’ disease. One of the largest studies that investigated the familial incidence of Perthes disease was conducted among 310 Scottish patients [44]. Of the 310 patients with a family history of the disease (246 male, 64 female; male to female sex ratio 3.8:1), 35 had bilateral disease (11.3%). Five dizygous and one monozygous twin pairs were present among the patients, with only one sibling affected in each of them. This study highlighted low overall proportions of affected relatives, with only a 1.6% frequency of appearance among siblings.

The largest study on twins explored the incidence of Perthes disease among 81 twin pairs of Danish children with at least one affected sibling [45]. Nevertheless, among the 10 pairs of monozygotic, 51 dizygotic and 20 unknown zygosity (UZ) twins described in this study, only 4 concordant pairs were found: 2 of each in the dizygotic and UZ groups. As expected, statistical analysis showed a low probability of a hereditary component of this disorder.

Interestingly, there are families that are prone to joint diseases similar to Perthes disease, such as avascular necrosis or secondary osteoarthritis. A common marker for this type of heritable susceptibility is collagen type II protein.

### 3.5. COL2A1 Mutations

Collagen type II is a fibrillary structural protein mainly found in hyaline and articular cartilage, but also in other tissues, such as the intervertebral disc (nucleus pulposus), retina (as a part of the clear gel that fills the eyeball (vitreous humor)), sclera and lens of the eye, nose, inner ear, and external ear [46]. As with other types of collagens, collagen type II is a triple helix consisting of three type α1(II) monomers. Fibrils of collagen type II are cross-linked in the viscous proteoglycan matrix with collagen type IX, forming a structure that provides the tissue with strength and compressibility that allows it to resist large deformations in shape, giving joints the ability to absorb shocks [47]. Mutations are dispersed throughout the entire gene, and no mutational hot spots within the *COL2A1* gene have been identified. Several publications reported the presence of missense mutations in the *COL2A1* gene (COL2A1 OMIM#120140) in Perthes patients in sporadic but also in familial cases [48,49,50,51]. All mutations found in Perthes patients are missense types of mutations leading to the substitution of the amino acid glycine with another, usually larger, amino acid, predominantly serine (Figure 1). Glycine is a small, nonpolar amino acid, and its substitution with a larger amino acid leads to structural changes in the triple-helical collagen type II structure [47].

The recurrent p.Gly1170Ser mutation (c.3665 G > A) in exon 50 of the *COL2A1* gene was first reported in two Taiwanese families with ANFH [52] and then in Japanese and Chinese families with Perthes disease [48,49]. Miyamoto et al. described three generations containing seven family members affected by Perthes disease, in whom the p.Gly1170Ser mutation was detected [49]. Another comprehensive genetic study was conducted in order to trace the same mutation through a five-generation Chinese family affected by three different joint diseases [48]. Of the forty-two family members analyzed in this study, sixteen members were affected by one of three joint diseases: Perthes disease (five affected members), avascular necrosis (AVN) of the femoral head (six affected members) or osteoarthritis (OA) (five affected members). Genetic analysis showed the presence of the c.3665 G > A mutation of the *COL2A1* gene in the heterozygous state in all affected family members, with the exception of one 8-year-old carrier, who was clinically asymptomatic. The authors concluded that this genetic variant was a disease-causing mutation in this family.

In another family study, a similar phenotypic presentation was described as a consequence of another mutation in the *COL2A1* gene, c.1888 G > A, p.Gly630Ser in exon 29 [51]. In order to detect this disease-causing mutation and trace its segregation, forty-five members of a four-generation Chinese family were recruited and analyzed. The study found that six family members were affected by either Perthes disease or ANFH and, in all of their DNA samples, the mutation c.1888 G > A of the *COL2A1* gene was detected. The same mutation was detected in a 7-year-old girl who was asymptomatic. The mutation was absent in samples from healthy family members and controls.

These two family studies are examples of the presence of a disease-causing mutation in the *COL2A1* gene, with a different onset of clinical manifestation. Factors that could influence the closure of the femoral head epiphysis, such as hormonal status, body weight or circulation problems, appear to be critical factors in determining phenotypic expression and defining the age of the onset of disease phenotypes.

#### COL2A1 Mutations in Patients with Bilateral Presentation

Mutations in the *COL2A1* gene were rarely found in sporadic cases [53]. The only study where mutations in the *COL2A1* gene were found in sporadic cases was the study by Kannu et al., which described two children with bilateral presentations of Perthes disease [50]. Two different mutations were found in probands (c.2014 G > T (p.Gly672Cys) and c.638 G > A (p.Gly213Asp)), but not in their family members. This very interesting finding highlights the need to seek COL2A1 mutations in bilateral cases of Perthes disease, which could be beneficial for differential diagnosis and management of patients.

### 3.6. Biological Therapy for Perthes Disease

The accumulation of data on particular pathophysiological changes at various stages of the disease makes biological therapy a promising option for the treatment of patients (Table 2). Bone turnover relies on biogenesis and the activity of OB and OC, and their number and function are tightly regulated. In this respect, the receptor activator of nuclear factor κ-Β ligand (RANKL) plays a fundamental role, and the fine-tuning of this process is controlled by its receptors: a transmembrane receptor for RANK and a soluble decoy receptor, osteoprotegerin (OPG) [24].

The mode of action of therapies that can target OC-mediated bone resorption and prevent femoral head deformity is elucidated. The fragmentation (resorptive) stage of Perthes disease lasts for 1–2 years [24] and agents that could attenuate bone resorption and stimulate bone formation, such as bisphosphonates (BP) and RANKL inhibitors, could be beneficial at this stage. Data about the usage of BP (Ibandronate and Zoledronic acid) from a pig model of ischemic necrosis [54], rat model of Perthes disease [55] and rat model of traumatic osteonecrosis [56] have shown that systemically administered BP has a protective role regarding the femoral head. The mode of action of these drugs is the stimulation of OC apoptosis, without affecting the inhibition of OC or new bone formation [57]. Nevertheless, BP have many limitations, especially when used with children. Their long half-life can have inhibitory effects on bone growth, and their systemic administration has limited access to the necrotic bone [57]. An ongoing clinical trial in Australia (ACTRN12610000407099) will compare the results from the intravenous administration of zolendronic acid in 100 children receiving the standard treatment [58].

However, subcutaneous injections of osteoprotegerin (human OPG-Fc) have been shown to have a protective role in the femoral head in a piglet model of Perthes disease [24]. Results indicate that RANKL inhibition produced a decreased total number of OC in repair tissue. Despite the finding that RANKL inhibitors do not bind to bone and that their effect is reversible since they are rapidly cleared from circulation, RANKL inhibition can be considered a novel therapeutic strategy for femoral healing deformity diseases.

#### 3.6.1. Anti-IL-6 Therapy

The influence of IL-6 on bone turnover and the effects of the blockade of the IL-6 receptor have been proven in in vitro and animal models and clinical studies of rheumatoid and juvenile arthritis [59,60,61,62]. Since the first confirmation of the role of IL-6 in Perthes disease [31,36], an in vitro study showed that the expression of IL-6 is HIF-1 dependent and is attenuated by the anti-IL-6 monoclonal antibody, tocilizumab [23]. These findings converged in the preclinical study of Ren et al., who used tocilizumab on a piglet model of Perthes disease to explore its influence on hip synovitis and bone healing [63]. Biweekly intravenous injections in doses of 15 to 20 mg/kg were applied in this preclinical setting and, after examination of the experimental and control groups, encouraging results were obtained. Significantly higher bone volume and a lower number of OC per bone surface were shown in the tocilizumab group compared to the control group. Additionally, the number of synovial macrophages and vessels was significantly lower in the tocilizumab group compared to the control group, showing that both the effect on attenuated osteoclastogenesis and the inhibition of angiogenesis had a positive influence on alleviated hip synovitis, supporting bone healing after treatment.

#### 3.6.2. Cellular Therapy as an Option for the Treatment of Perthes Disease

Constant improvements in understanding the biology and course of action of mesenchymal stem cells (MSCs) have encouraged their application and modernized the field of skeletal regenerative medicine. Pediatric cases of bone and joint diseases could potentially have enormous benefits for treatment using stem cells. Severe congenital impairments of bone metabolism, such as osteogenesis imperfecta (OI), have already been successfully treated using MSCs prenatally [64,65] and postnatally [66]. MSCs could be a promising option for diseases where no other treatment options are available and for chronic diseases that require lifelong therapy or where standard therapy gives unsatisfactory results.

Usage of MSCs in the pediatric population must be preceded by sufficient and encouraging results from preclinical trials and safety and efficacy clinical studies and adequate results from treatments of similar diseases in adults. A pioneering attempt in the treatment of Perthes disease using stem cells was described in a study by Wang et al., where the authors used a combination of drilling through the growth plate with injections of adipose-derived mesenchymal stem cells (ADSC) and bone morphogenetic protein 2 (BMP-2) on an ischemic model of epiphyseal ischemic necrosis of the femoral head in juvenile rabbits [67]. The treatment showed new bone formation and prevented the collapse of the femoral head epiphysis in an experimental model of Perthes disease.

There are also available results from the treatment of other pediatric joint diseases, such as juvenile idiopathic arthritis (JIA), which is a form of juvenile rheumatoid arthritis. Ten patients who were treated using umbilical cord-derived MSCs were followed for 2 years. No side effects were detected. The patients’ symptoms improved and the dose of antirheumatic drugs decreased during the follow-up period with good tolerance [68]. This is an important result, suggesting a new paradigm of treatment for this severe condition.

## 4. Conclusions

The delineation of molecular biomarkers that underlie the pathophysiological process of Perthes disease allows for the possibility to provide earlier and more accurate diagnoses and more precise follow-ups and treatment in the early phases of the disease. In this respect, techniques that analyze extracellular particles from the patient’s plasma could be used as a new tool for early diagnosis and follow-up. Markers involved in endothelial cell dysfunction; markers of angiogenesis, inflammation and apoptosis; and genes involved in bone turnover are possible new targets for biological therapy. In the years to come, high throughput methods of sequencing, such as whole exome sequencing (WES) and whole genome sequencing (WGS), will provide additional data on other genetic loci that could be of interest for the development of Perthes disease.

## Figures and Tables

**Figure 1 diagnostics-13-00471-f001:**
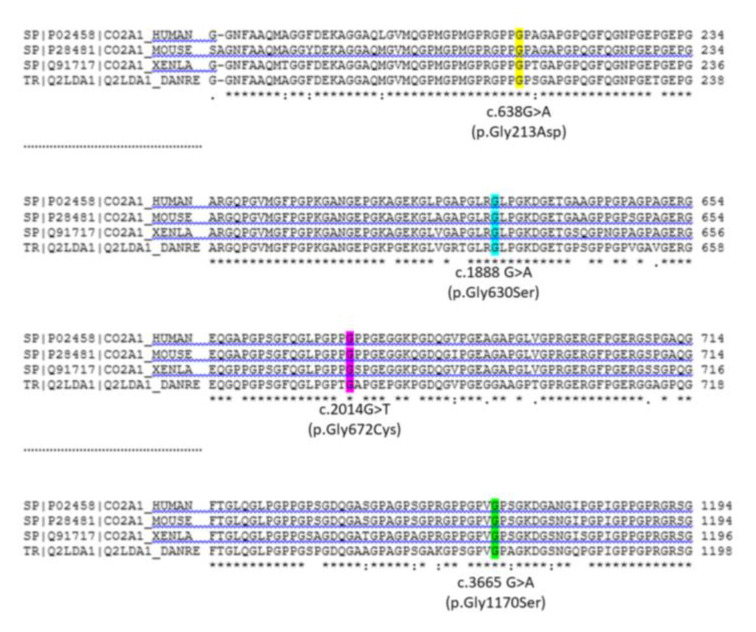
Protein sequence alignment of COL2A1 protein. COL2A1 proteins show high conservation of the primary structure among distant species. The figure shows CLUSTAL OMEGA (1.2.4) multiple sequence alignment of the regions of COL2A1 protein of different organisms: Human, Mus musculus, Xenopus laevis, and Danio rerio [Uniprot identifiers: P02458, P28481, Q91717, Q2LDA, respectively]. Highlighted amino acids show the mutations found in Perthes patients. c.638 G > A (p.Gly213Asp) (exon 9); c.2014 G > T (p.Gly672Cys) (exon 31) (NM_001844.5) [50]; c.1888 G > A (p.Gly630Ser) (exon 29) (NM_033150.3) [51]; c.3665 G > A (p.Gly1170Ser) (exon 50) (NM_033150.3) [48,49]. The alignment of the whole protein sequence is given in Appendix A.

**Table 1 diagnostics-13-00471-t001:** Studies on biomarkers in Perthes disease.

Biomarker	SNV	Experimental Model	Reference
*VEGF*		piglet model	Kim et al. [12]
*VEGF*		rabbit model	Li et al. [13]
*VEGF*		human serum	Sezgin et al. [14]
*VEGF*		human serum	Tiwari et al. [15]
*eNOS*	894 G > T 4a4b	human peripheral blood	Zhao et al. [19]
*eNOS*	−786 T > C; 894 G > T 4a4b	human peripheral blood	Azarpira et al. [20]
*EMPs vesicles*		human plasma	Li et al. [21]
*exosomes*		human plasma	Huang et al. [22]
*IL-6* *TLR4*	174 G > C; −597 G > A D299G; T399I	human peripheral blood	Srzentic et al. [31]
*TNF-α* *IL-3*	−308 G > A −16 C > T	human peripheral blood	Srzentic et al. [32]
*IL-6*	−174 G > C; −572 G > C	human peripheral blood	Akbarian-Bafghi et al. [35]
*IL-6*		human synovial fluid	Kamiya et al. [36]
*IL-6*		piglet model	Yamaguchi et al. [23]
*IL-6*		murine model	Kuroyanagi et al. [37]
*TLR4*		piglet model	Adapala et al. [38]
*BAX*, *BCL-2*		human peripheral blood	Srzentic et al. [32]
*Bax*, *miR-214*		human cartilage, serum and chondrocytes	Zhu et al. [42]
*COL2A1*	c.3665 G > A	human peripheral blood	Su et al. [48]
*COL2A1*	c.3665 G > A	human peripheral blood	Miyamoto et al. [49]
*COL2A1*	c.638 G > A c.2014 G > T	human peripheral blood	Kannu et al. [50]
*COL2A1*	c.1888 G > A	human peripheral blood	Li et al. [51]

**Table 2 diagnostics-13-00471-t002:** Biological treatments of Perthes disease.

Therapy	Experimental Model	Reference
Osteoprotegerin	piglet model	Kim et al. [24]
Ibandronate	pig model	Kim et al. [54]
Zoledronic acid	rat model	Little et al. [55]
Zoledronic acid	rat model	Little et al. [56]
Tocilizumab	piglet model	Ren et al. [63]
ADSC	rabbit model	Wang et al. [67]

## Data Availability

The data presented in this study are publicly available on PubMed (https://pubmed.ncbi.nlm.nih.gov/ (accessed on 1 October 2022)), and in Appendix A.

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
