# Peer review of "Molecular Biomarkers in Perthes Disease: A Review"

_diagnostics, 2023, doi:10.3390/diagnostics13030471_

Round 1
Reviewer 1 Report
See attached review.

Author Response
We are very thankful for this valuable review. We carefully acknowledged all suggestions and introduced changes accordingly within the manuscript. We believe that it is now improved and much more understandable. Concerning the quality of English language, we consulted native English speaker, and all changes are introduced with “Track changes” option so to be easily visible for the reviewers and editors.

Reviewer 2 Report
This study entitled “Molecular biomarkers in Perthes disease: From molecular markers to therapeutic targets” seems to have been generally well executed and written. Furthermore, I believe that this research will be of great interest to the readers. Finally, I have only a few minor suggestions to improve the quality of the paper.
Title
Please add the type of your study in the title, and also your title could be improved so please rewrite it.
Materials and Methods
2.1. Literature search strategy and selection criteria
In your first sentence of this section “Inclusion criteria for this review” please add the type of review that you have performed (e.g., systematic,…). If you executed a systematic review, was it written according to PRISMA statement (please state this in the text).
Please provide the initials of authors who performed the screen of available literature, and the initials of author who solved any disagreements among authors and approved the final list of included studies.
Please be a more specific about time range of your study (use day/month/year format i.e., DD/MM/YYYY, not just a “date was set from 1992 to 2022”).
The final minor remark of your paper in my opinion, is that your searched only for PubMed, so please state this issue as limitation of your work.
Author Response

(The authors gave the same response as above.)

Round 2
Reviewer 1 Report
Changing the manuscript to a narrative review improves the robustness of the paper.
I would suggest keeping the original search algoritm as an appendix.